
# Brief communication: simple-INSYDE, development of a new tool for flood damage evaluation from an existing synthetic model

Marta Galliani, Daniela Molinari, and Francesco Ballio

Department of Civil and Environmental Engineering, Politecnico di Milano, Milan, 20133, Italy

*Correspondence to*: marta.galliani@polimi.it

**Abstract**

INSYDE is a multi-variable, synthetic model for flood damage assessment to dwellings. The analysis and use of this model highlighted some weaknesses, linked to its complexity, that can undermine its usability and correct implementation. This study proposes a simplified version of INSYDE which maintains its multi-variable and synthetic nature, but has simpler

mathematical formulations permitting an easier use and a direct analysis of the relation between damage and its explanatory variables.

## 1   Introduction

INSYDE (IN-depth SYnthetic Model for Flood Damage Estimation, Dottori et al., 2016) is a synthetic model for the estimation of flood damage to residential buildings at the micro scale (i.e. building level), developed and tested in Italian case studies

(Amadio et al., 2019, Molinari and Scorzini, 2017, Molinari et al., 2020). The monetary damage to a dwelling is computed in the model as the sum of 33 different components, referring to the costs of reparation, removal and replacement of the damaged elements, which are functions of several damage explicative variables, related both to the hazard and to the vulnerability of the affected item (Table 1). Since the same explicative variable may directly or indirectly influence more than one damage component, it is difficult to understand the weight that each explicative variable has on the overall damage estimate. Moreover,

the complex and articulate structure of INSYDE could discourage the implementation of the model and its use through other platform such as GIS software. This study proposes an alternative version of the model, named simple-INSYDE, which aims at overcoming these limitations. Simple-INSYDE preserves the multi-variable nature of the model, but aggregates damage components in a smaller set of functions, which clearly describe the role of each explicative variable on the total damage figure and can be easily implemented, even by non-expert users. Such functions are calibrated for low-velocity floods, with building

characteristics typical of Northern Italy. The method and the assumptions implemented to obtain the simplified version of the model are described in the following sections.




**Table 1**. Input variables considered in INSYDE (Dottori et al., 2016).

| | Event features and building characteristics variables in INSYDE | Default values in INSYDE | Independent variables chosen in simple-INSYDE |
|---|---|---|---|
| $h_e$ | water depth outside the building [m] | [0;5] Incremental step: 0.01m | x |
| h | water depth inside the building [m] | h = f ($h_e$, GL) | Fixed at default value |
| v | maximum velocity of the water perpendicular to the building [ms$^{-1}$] | 0.5 | Fixed at default value |
| s | sediment load [% on the water volume] | 0.05 | Fixed at default value |
| du | duration of the flood event [h] | 24 | x |
| q | water quality (presence of pollutants) | Yes | x |
| FA | Footprint Area [m$^2$] | 100 | x |
| IA | Internal Area [m$^2$] | 0.9 FA | Fixed at default value |
| BA | Basement Area [m$^2$] | 0.5 FA | x |
| EP | External Perimeter [m] | 4√FA | Fixed at default value |
| IP | Internal Perimeter [m] | 2.5 EP | Fixed at default value |
| BP | Basement Perimeter [m] | 4√BA | Fixed at default value |
| NF | Number of floors | 2 | Functions for one storey |
| IH | Interfloor height [m] | 3.5 | Fixed at default value |
| BH | Basement height [m] | 3.2 | Fixed at default value |
| GL | Ground floor level [m] | 0.1 | Fixed at default value |
| BL | Basement level [m] | – GL – BH – 0.3 | Fixed at default value |
| BT | Building type (1 detached house, 2 semi-detached, 3 apartment) | 1 | Fixed at default value |
| BS | Building structure (1 reinforced concrete, 2 masonry) | 2 | x |
| FL | Finishing level (1.2 high, 1 medium, 0.8 low) | 1.2 | x |
| LM | Level of maintenance (1.1 high, 1 medium, 0.9 low) | 1 | x |
| YY | Year of construction | 1994 | Fixed at default value. |
| PD | Heating system distribution (1 centralized, 2 distributed) | 1 if YY≤1990, 2 otherwise | Fixed as 1 (centralized) |
| PT | Heating system type (1 radiator, 2 pavement) | 2 if YY>2000 and FL>1, 1 otherwise | Fixed as 1 (radiator) |






## 2 Method

The first step to provide a simpler structure of the model was to aggregate the original damage functions into four components:

- Damage to basement: in case of flood, basement is assumed totally inundated and damage does not depend on water level.

- Damage to floor: in case of water level higher than the level of floor, the damage to floor is counted as independent from water level.

- Damage to storey: it considers damage to the elements over the floor (e.g., walls and plants) that depends on water level.

- Damage to boiler: it depends on water level only if the basement is not present, otherwise, the boiler is considered located in the basement which is completely inundated.

In order to support model transferability (Merz et al., 2010), the simplified model computes damage in relative terms, as the
ratio of the absolute damage to a reference value. The reference value is set as the cost of reconstruction of the storeys exposed to the flood; cost of reconstruction is evaluated as the product of the replacement value RV [€/m$^2$] and the footprint area A of each storey [m$^2$]. Equation (1) represents the conceptual formula of the simplified model, where $D$ is the building damage in absolute term [€], $d$ in relative term, $n$ is the number of flood exposed storeys.

$$D = RV_{basem} \cdot A_{basem} \cdot d_{basem} + RV_{storey} \cdot A \cdot \left( \sum_{i=1}^{n} (d_{storey_i} + d_{floor_i}) + d_{boiler} \right) \tag{1}$$

The second step was the choice of the independent variables to be included in the model, among those of the original INSYDE (Table 1). The variables that were not included in simple-INSYDE were not effectively neglected, but implicitly assumed at the default values according to the assumptions made in INSYDE, for the geographical context and the flood type of interest (Wagenaar et al., 2016). Among the event feature variables, we preserved the water level, the duration of the flood and the presence of pollutants. Indeed, the sensitivity analysis performed in Dottori et al. (2016) highlighted that, in case of slow
riverine flood events, water velocity and sediment load have a minor influence on damage, compared to the chosen variables. The selection of the vulnerability variables followed different criteria. We considered the interfloor height and the basement height fixed at their default values because they do not vary significantly in Northern Italy. We kept the default value also for the ground floor level and the heating system variables (PD and PT), because information on them is difficult to retrieve, without a detailed field survey. The internal area, the external perimeter, the internal perimeter, the basement perimeter and
the basement level are fixed as functions of other variables in INSYDE, and we maintained this assumption. However, this implies limiting the use of the model for the estimation of damage to single housing units, and not to condominiums; indeed, the functions to estimate perimeters in INSYDE were established considering the typical configuration of a 100 m$^2$ detached Italian house; this configuration is kept constant in the model, thus not considering a variation in the number of rooms or a multiplication of housing units in case the building area increases. The remaining vulnerability variables were the object of a
sensitivity analysis, which revealed that the year of construction and the building type do not significantly affect damage estimate. On the other hand, the building type, in Italy, is important to evaluate the replacement value. Table 1 shows the variables that were finally considered in simple-INSYDE.



The last step was the development of the simplified functions. For the four damage components, a set of reference values was
defined for each variable that influences the component under investigation. Then, the damage component was computed by
varying its input variables one by one around the reference value, in order to identify simple interpolating functions suitable
for representing the role of each variable on the final damage figure. Results are expressed by equations 2-5:

$$d_{basement} = f(A_{basem})f(du) \rightarrow \begin{cases} f(A_{basem}) = 0.02 + \frac{0.35}{\sqrt{A_{basem}}} \\ f(du) = 1 + 0.3 \arctan(du - 36) \end{cases} \quad (2)$$

$$d_{storey} = f(h)f(A)f(LM,du)f(BS)f(FL)f(q) \rightarrow \begin{cases} f(h) = \left(0.17h - 0.02h^2\right) \\ f(A) = \left(0.2 + \frac{7}{\sqrt{A}}\right) \\ f(LM,du) = \begin{cases} 1 + 0.15 \cdot \arctan(du - 36) & \text{if LM low} \\ 0.8 + 0.2 \cdot \arctan(du - 36) & \text{if LM high} \end{cases} \\ f(BS) = \begin{cases} 1.35, & \text{if BS masonry} \\ 1, & \text{elsewhere} \end{cases} \\ f(FL) = \begin{cases} 1.5, & \text{if FL high} \\ 1, & \text{elsewhere} \end{cases} \\ f(q) = \begin{cases} 1.2, & \text{if q = 1, presence of pollutants} \\ 1, & \text{elsewhere} \end{cases} \end{cases} \quad (3)$$

$$d_{floor} = f(h, FL) = \begin{cases} 0.04, & \text{if h>0 and FL high} \\ 0, & \text{elsewhere} \end{cases} \quad (4)$$

$$d_{boiler} = f(A_{basem}, h) = \begin{cases} 0.015, & \text{if } A_{basem} \neq 0 \text{ or } A_{basem} = 0 \text{ and } h > 1.6 \text{ m} \\ 0, & \text{elsewhere} \end{cases} \quad (5)$$

where the units of measures of the variables are m$^2$ for the area (A), hours for duration (du), and m for water depth (h).
The interpolating functions were calibrated comparing the damage simulated by the simplified model and the original model,
for a sample of 10000 buildings, whose features (i.e. input variables values) were randomly selected from probability
distributions assumed representative of Northern Italy (Table 2). In particular, for the footprint area, the finishing level, the
building structure and the maintenance level, the distribution parameters were chosen on the bases of real estate data of
Northern Italy. The comparison of the simulated damage by the original and the simplified model, showed a mean relative
error equal to 0.24, with a ratio to the mean absolute damage equal to 1.07. The application of the model INSYDE in real case
studies (Dottori et al., 2016, Molinari and Scorzini, 2017, Amadio et al., 2019, Molinari et al., 2020) showed good performance
of the model, with a mean ratio between the total damage simulated and the observed damage equal to 1.26. On the other hand,
literature shows that the performance of flood damage models can be affected by high uncertainty, with relative errors vary
from 20% to exceed 1000% (Scorzini and Frank, 2017, Thieken et al., 2008). Thus, we consider that the additional error caused
by the use of simple-INSYDE is acceptable, and that the estimation of the overall damage is comparable with that supplied by
INSYDE.





**Table 2.** Probability distributions and respective parameters of the explicative variables.

| Variable | Distribution | Parameters | Notes |
|---|---|---|---|
| A | Log-Normal | $\mu=5.10$, $\sigma^2=0.49$ | $\mu$ mean, $\sigma^2$ variance |
| FL | Bernoulli | p=0.02 | probability FL high |
| BS | Bernoulli | p=0.64 | probability BS masonry |
| LM | Bernoulli | p=0.86 | probability LM high |
| NF | Discrete Uniform | [1,10] | |
| Basement | Discrete Uniform | [0,1] | 0 absent, 1 present |
| BT | Discrete Uniform | [1, 3] | 1 detached house, 2 semi-detached, 3 apartment |
| du | Discrete Uniform | [10, 60] | unit of measure: hour |
| h | Continuous Uniform | [0, 3.5] | unit of measure: meters |
| q | Discrete Uniform | [0, 1] | 0 absent, 1 present |

### 3 Discussion

This study led to the main objective of developing a new tool for flood damage estimation to dwellings, which is based on a sensible number of available input data, and allows investigating the relation between damage and its explanatory variables by means of a simple set of functions. For instance, Figure 1 shows the relative damage computed by simple-INSYDE as a function of water depth, for the different damage components of the model. The figure highlights that the storey component gives the biggest contribution to damage and is the only one depending on water level. The other components are independent

of water level and have a lower weight on the final damage figure, but they assume a non-negligible role, especially in case of shallow waters.

Moreover, the study allowed to deeply investigate the behaviour of the original model and to highlight shortcomings that could be further improved in the future. For example, assumptions made in the model on building configuration, which limit its use to single housing units and not condominiums, is not directly reported in the paper of Dottori et al. (2016), but is important for

a correct implementation of the model and a better understanding of estimation errors.

Compared to the original model, the simplified model requires fewer input variables, facilitating the model implementation, but impeding the control by the user on the variables that are implicitly considered. For this reason, Simple-INSYDE is less adaptable to contexts different from the calibration one than INSYDE. It is worth recalling that simple-INSYDE is addressed to evaluate damage in case of low-velocity floods and built environments typical of Northern Italy. It is recommended not to

use it for other types of inundation (Kreibich and Dimitrova, 2010) or for other types of building and/or geographical contexts. In these cases, the derivation of new interpolating functions from INSYDE, with the process described in this study is suggested; to this aim, the original model needs be adapted to the context of interest, by modifying the default values of the variables and the unit prices of the building components, then, the simplification method can be implemented to obtain new functions with new coefficients.



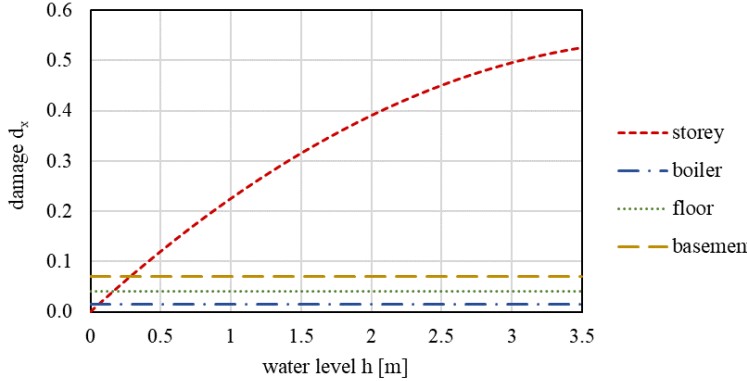


**Figure 1 -** Comparison of the simple-INSYDE damage components as functions of water depth. Damage values in the figure refer to the case of a 100 m$^2$ building, with cellar, in reinforced concrete, with high finishing level and low maintenance level, affected by a flood of 36 hours, in absence of pollution.

### Acknowledgments

Authors acknowledge with gratitude Anna Rita Scorzini, from the Department of Civil, Environmental and Architectural

Engineering, University of L'Aquila, for her fruitful suggestions and hints during the developing of the work.

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
