# Peer review of "Brief communication: simple-INSYDE, development of a new tool for flood damage evaluation from an existing synthetic model"

_Natural Hazards and Earth System Sciences, 2020_

## Referee Comment (RC1) · Anonymous Referee #1 · 14 May 2020

The manuscript describes a method for reducing complexity of a flood vulnerability model (INSYDE) by means of a sensitivity analysis. The result is a surrogate model that aims at enhancing the applicability of the model. The manuscript provides interesting results, also beyond the ones discussed. I would suggest to add a few sentences in the discussion section on the general added value of the paper (beyond the exclusive applicability in Italy), namely that complexity reduction or the development of surrogate models can be recommended without losing too much accuracy. This is especially the case, if not all of the independent variables are available as model input. Another suggestion is to add a reference to the dataset you mentioned in lines 87-88, the "real estate data of Northern Italy".

---

## Referee Comment (RC2) · Anonymous Referee #2 · 28 May 2020

The paper presents a simplified INSYDE model by reducing the number of damage predictor variables and treating the eliminated predictors as constants in the equations. The authors perform sensitivity analysis to identify the important predictors. From the manuscript, the objectives and the approach are structured and clear. I understand that INSYDE (Dottori et al. 2016) is designed to consider missing/unavailable input data. The default values which go in the place of missing values are based on flood affected regions in Northern Italy. Since the sensitivity analysis for hazard related predictors have already been performed (Dottori et al. 2016), the sensitivity analysis for other predictors and the altered simple-INSYDE equations are potential novel aspects of this study. Some points to improve the manuscript concerning discussions

and applicability of the simple-INSYDE model. 1. The sensitivity analysis is not very clear. What is the basis for choosing the predictors? Also, does the model performance becomes worse when one/some of the chosen variables are treated as default values? I think, a clearer explanation will help to extend this approach to other synthetic models. Not many synthetic models allow missing variables. So, some details on how to choose the default values may be helpful. 2. Transferability appears to be an advantage of INSYDE (Dottori et al. 2016, Amadio et al., 2019). I agree with the authors that there is a need to reduce complexity in order to make the model widely applicable. But, feeding in more of local inputs as default values may cause serious bias when the model is transferred, as such. The authors mention about this. But, an illustration of how to work around this limitation will add value to the study. 3. It is also not clear how the calibration of simple-INSYDE is implemented (line: 84). Is the INSYDE original model applied with the same set of variables (Table 2) considering the rest as missing/default and then an interpolation is performed or are there additional variables involved? 4. Since both the INSYDE model development and the simplification approach are based on same regions (Northern Italy), it is difficult to judge the general applicability of this simplification approach based on the reported errors. The comparison with same set of simulated buildings appears like evaluating the fit of the interpolation function with same train and test data. An alternative is that the authors may consider providing validation on real damage data like Amadio et al (2019). 5. Since INSYDE is a probabilistic model, some discussions on uncertainty in predictions from both INSYDE and simple-INSYDE will be an interesting addition to the discussions. Minor Comments 1. Some insights into 'why' INSYDE is complex and hard to implement (what aspects?) may add value to the objective. 2. Line 16: reparation may be replaced with repair 3. Line 21: How does this improvement help integration to a GIS software? This is not discussed. 4. Lines 21-24: Reduction in dimensionality of the INSYDE model should be included. The model doesn't completely preserve the multi-variable nature of INSYDE. Some variables are treated as constants in the simple model. 5. In Table 1, it is difficult to understand X. From the

context, I understood that these are values user has to input. A note will help. 6. Line 45: walls and plants? I think it is a typo. 7. Line 51: Footprint Area is interchangeably used as FA and A (table 1 and 2) 8. Line 70: There is no quantification provided for sensitivity analysis. Hence, the context for 'significantly' is missing. 9. Lines 91-94: The range of acceptable errors is very huge. Given this argument, even the need for important variables considered in simple-INSYDE may be questioned 10. Table 2 needs reference. Also, please introduce a column with full-forms to make it easy for the reader. 11. Lines 100: Please rephrase that simple-INSYDE is a simplified version of INSYDE. The fundamental assumptions and methodologies are from INSYDE. 12. The arrangement of the Discussion section 3 is not coherent. The model is for Northern Italy. But, more focus on wider applicability of such an approach and how to implement this for other regions will be interesting.

Please also note the supplement to this comment:
https://www.nat-hazards-earth-syst-sci-discuss.net/nhess-2020-76/nhess-2020-76-RC2-supplement.pdf

---

## Referee Comment (RC3) · Anonymous Referee #3 · 1 Jun 2020

The manuscript describes a new and simpler version of a the flood damage model INSYDE, recently proposed by Dottori et al. (2016). The content of the document and the model will be of interest for the community of the Journal.

Hereafter some comments that I hope could be helpful to the Authors to further improve the manuscript.

- Introduction: I guess one of the limitation of the original version of the model (INSYDE) might be related to availability of all required data. If this is the case, it could be worth mentioning it in the text. Also, please provide more details regarding difficulties with GIS application.

[Figure]

- Introduction, L19-21: the presentation of the limitations that inspired the new version is quite general. Can you better specify them?

- Table 1: table caption should also include the simple-INSYDE. Also, referring to simple-INSYDE, it might be helpful distinguishing independent hazardous variables respect to variable representative of the exposure. Please, specify the meaning of x used in table 1.

- Some of the variables assumed with a fix value in simple-INSYDE were constant also in the original model (see e.g. IH, BH, GL). What is the difference compare to INSYDE? What are the fixed values adopted for simple-INSYDE?

- Apart from a contained complexity, are there advantages on using a fixed values for same variables that can be directly estimated from other required in any case by the simple version? I am thinking for example to IA or EP, which depend on FA that is still necessary for the application of simple-INSYDE. Also in this case, how did you define those default values? Are they based on observations or assumptions?

L51: footprint area in table 1 is indicated as FA, not A. Also check Table 2.

L53: wouldn't be more correct saying "flood affected storeys"?

L55: wouldn't be more precise saying "independent hazardous variables"?

L62: I suggest providing more details on these values. This would help the possibility to, eventually, apply the model elsewhere.

L73-76: honestly, these steps are not clear to me. If you can extend the explanation, or provide an example, it would be much easier to understand the procedure.

Eq. 3): what happen in case LM is medium?

Table 2: in order to make the model transferability easier it would be worth reporting in table 2 the range of the values considered for all the variables.

[Figure]

---

## Author Response (AR1)

**Response to the reviews**

We would like to thank the Referees for their careful and interesting evaluation of this Brief Communication; their remarks and advice will help us to make the paper more clear, readable, and complete. In the following, a point by point answer to questions and comments raised by the referees is supplied.

The numbers of lines outside the brackets refer to the marked-up version of the manuscript, in brackets to the new version.

**Referee #1**

*R1-C1: I would suggest to add a few sentences in the discussion section on the general added value of the paper (beyond the exclusive applicability in Italy), namely that complexity reduction or the development of surrogate models can be recommended without losing too much accuracy. This is especially the case, if not all of the independent variables are available as model input.*

Thank you for the suggestion, we agree that highlighting the strength of the work, also as a method to develop surrogate models without losing too much accuracy, could increase its value. We added a sentence in the discussion section to underline this concept, lines 159-160 (128-129).

*R1-C2: Another suggestion is to add a reference to the dataset you mentioned in lines 87-88, the "real estate data of Northern Italy".*

Thank you for the comment. In fact, "real estate data of Northern Italy" could be misleading, as the reader could think that there is a unique database with all the information. But the types of probability distributions and their parameters were chosen on the basis of information coming from different databases. For instance, the percentage of buildings with high or low level of maintenance was computed according to the provincial data of Istat (Italian institute of statistics) for Piemonte, Lombardia, Emilia Romagna and Veneto region; for the estimation of the percentage of building with high or low finishing level, we associated the finishing level to the cadastral categories supplies by the real estate market observatory of revenue agency (Osservatorio del Mercato Immobiliare OMI by Agenzia delle Entrate) in the same regions. In the paper, we will be more precise about the source of information, lines 92-95 (85-88). However, if the method wants to be repeated, different probability distributions and information sources can be used, according to the territorial context and available databases.

**Referee #2**

*R2-C1: The sensitivity analysis is not very clear. What is the basis for choosing the predictors? Also, does the model performance becomes worse when one/some of the chosen variables are treated as default values? I think, a clearer explanation will help to extend this approach to other synthetic models. Not many synthetic models allow missing variables. So, some details on how to choose the default values may be helpful.*

We chose the predictors and the implicit variables according to different criteria, explained in lines 59-71 (55-67). For instance, we neglected variables that are not available or considered fixed or that are functions of other variables. We implemented a local sensitivity analysis consisting in perturbing one single parameter around its reference value, keeping the others constant, to evaluate the resulting variation of the output. The parameters that do not cause significant changes in damage estimation were neglected. In particular, the variables that were finally included in simple-INSYDE cause a variation of damage between 12% and 38%, while neglected ones less than 2%. In the original version of the paper, we did not describe the sensitivity analysis in detail because of the brief and concise nature of the manuscript, but we understand that is too synthetic; in the new version we added some information without going into details, lines 77-79 (71-74). Still, it is worth noting that the implemented sensitivity analysis is one possible method to select relevant variables; other modellers may choose other procedures they consider most appropriate.

Regarding the model performance, we did not evaluate it in the case also the chosen explicative variables are set as implicit (e.g. in the case the model became bi or uni-variable), but the simple structure of the model allows to easily implement this type of analysis, lines 131-132 (113-114).

*R2-C2: Transferability appears to be an advantage of INSYDE (Dottori et al. 2016, Amadio et al., 2019). I agree with the authors that there is a need to reduce complexity in order to make the model widely applicable. But, feeding in more of local inputs as default values may cause serious bias when the model is transferred, as such. The authors mention about this. But, an illustration of how to work around this limitation will add value to the study.*

As explained in the discussion chapter, we think that the simplification of INSYDE could make the model widely applicable in the context of calibration, Northern Italy, but it does not make the model more transferable in other contexts. To transfer the model, it is necessary to adapt the original INSYDE to the context of interest, as mentioned in Dottori et al. (2016), and then to apply the procedure shown in the paper with assumptions coherent with the development context.

*R2-C3: It is also not clear how the calibration of simple-INSYDE is implemented (line: 84). Is the INSYDE original model applied with the same set of variables (Table 2) considering the rest as missing/default and then an interpolation is performed or are there additional variables involved?*

Yes, the variables that are not in Table 2 are considered as default. The model INSYDE and simple-INSYDE work with the same set of buildings and flood features. The values of the functions' coefficients were manually changed to reduce the error. We modified the part of calibration in order to make clearer the procedure, lines 89-92 (82-85).

*R2-C4: Since both the INSYDE model development and the simplification approach are based on same regions (Northern Italy), it is difficult to judge the general applicability of this simplification approach based on the reported errors. The comparison with same set of simulated buildings appears like evaluating the fit of the interpolation function with same train and test data. An alternative is that the authors may consider providing validation on real damage data like Amadio et al (2019).*

Thank you for the comment. The aim of the comparison was to evaluate the error if we use simple-INSYDE instead of INSYDE, for this reason we used the same set of building parameters and flood parameters. We assumed that the performance of the original INSYDE is good in the panorama of Italian flood damage models, without providing an evaluation as it has been already done in other papers quoted in the introduction: e.g. Amadio et al., 2019, Molinari and Scorzini, 2017, Molinari et al., 2020.

*R2-C5: Since INSYDE is a probabilistic model, some discussions on uncertainty in predictions from both INSYDE and simple-INSYDE will be an interesting addition to the discussions.*

INSYDE is a probabilistic model because some damage mechanisms are modelled using probabilistic functions rather than deterministic (Dottori et al., 2016). Simple-INSYDE was developed through deterministic interpolations of the model functions. The investigation of how uncertainty in damage predictions propagates into the output is interesting but out of the scope of the paper.

*R2-C6: 1. Some insights into 'why' INSYDE is complex and hard to implement (what aspects?) may add value to the objective.*

Thank you for the suggestion. The main feature behind INSYDE complexity is its articulate structure, made of a lot of functions. This fragmented structure prevents to understand the relation between damage and its explanatory variables. Moreover, the model is now available in R. Simpler functions could be implemented and understood also by users that do not know R or that want to integrate the damage functions with other analysis tools. We added a sentence to clarify this aspect, lines 20-21 (19-21).

*R2-C7: 2. Line 16: reparation may be replaced with repair*

Yes, thank you for the advice.

*R2-C8: 3. Line 21: How does this improvement help integration to a GIS software? This is not discussed anywhere.*

The reference to GIS software was an example, but we see that it is not clear. The idea is that simpler functions are simpler to be implemented with other programming language or to be integrated with other tools, as the GIS tool. We changed this sentence, lines 20-21 (19-21).

*R2-C9: 4. Lines 21-24: Reduction in dimensionality of the INSYDE model should be included. The model doesn't completely preserve the multi-variable nature of INSYDE. Some variables are treated as constants in the simple model.*

Yes, simple-INSYDE considers less variables than INSYDE, but its nature remains multi-variable because it considers several damage explanatory variables and represents damage mechanisms that are functions of a lot of variables, even if some parameters are implicitly considered.

*R2-C10: 5. In Table 1, it is difficult to understand X. From the context, I understood that these are values user has to input. A note will help.*

Thank you, "x" is not clear, we added a note with x meaning in Table 1.

*R2-C11: 6. Line 45: walls and plants? I think it is a typo.*

Thank you, we replaced plants with systems (we mean electrical or heating system).

*R2-C12: 7. Line 51: Footprint Area is interchangeably used as FA and A (table 1 and 2)*

Yes, we used A and not FA for the simplified model. We added a note in Table 1.

*R2-C13: 8. Line 70: There is no quantification provided for sensitivity analysis. Hence, the context for 'significantly' is missing.*

As said in R2-C1, we did not describe in detail the sensitivity analysis because of the short nature of the paper and because it is a possible way to select the variables, but not the unique. However, we enriched the description, also providing some quantitative performance indexes, lines 77-79 (71-74).

*R2-C14: 9. Lines 91-94: The range of acceptable errors is very huge. Given this argument, even the need for important variables considered in simple-INSYDE may be questioned*

Yes, the range is huge, but it refers to an example from the application of the FLEMO-ps model (Thieken et al., 2008); the example was chosen to emphasize that uncertainty in flood damage estimation could be very high. The discussion about the choice of considering very few or several variables, so to discuss advantages and disadvantages of uni- or multi-variable models, is interesting but it needs further analysis and discussion that transcends the objective of this paper. According to the choice of a brief communication, we prefer focusing on the main objective of the study.

*R2-C15: 10. Table 2 needs reference. Also, please introduce a column with full-forms to make it easy for the reader.*

There is not a specific reference, we choose the value of probability distributions according to data about building characteristics and real estate market in some regions of Northern Italy (see R1-C2). We changed the sentence about the calibration and Table 2 in order to be more precise about the data sources and the probability distributions, but we prefer to use the synthetic notation to reduce space and because the full-forms are easily available in statistical books or websites.

*R2-C16: 11. Lines 100: Please rephrase that simple-INSYDE is a simplified version of INSYDE. The fundamental assumptions and methodologies are from INSYDE.*

Ok thank you, we rephrased it.

*R2-C17: 12. The arrangement of the Discussion section 3 is not coherent. The model is for Northern Italy. But, more focus on wider applicability of such an approach and how to implement this for other regions will be interesting.*

See answer R2-C2.

**Referee #3**

*R3-C1: Introduction: I guess one of the limitation of the original version of the model (INSYDE) might be related to availability of all required data. If this is the case, it could be worth mentioning it in the text. Also, please provide more details regarding difficulties with GIS application.*

About the obstacle linked to the availability of the required data, the original INSYDE overcomes this by setting default values for missing variables. About the GIS application, it was an example, but we see that is not clear (see also answer R2-C8). The point is that simpler functions are simpler to be implemented with other programming language or to be integrated with other tools, as the GIS tool. We changed this sentence to clarify the concept, lines 20-21 (19-21).

*R3-C2: Introduction, L19-21: the presentation of the limitations that inspired the new version is quite general. Can you better specify them?*

The main reason that inspired the new version is the articulated structure of the INSYDE model. We think that a simpler structure could facilitate the analysis of the role of the variables in damage computation and the easily implementation of the model. See also R2-C6.

*R3-C3: Table 1: table caption should also include the simple-INSYDE. Also, referring to simple-INSYDE, it might be helpful distinguishing independent hazardous variables respect to variable representative of the exposure. Please, specify the meaning of x used in table 1.*

Yes, thank you for the advice, we adjusted the table and added the meaning of x.

*R3-C4: Some of the variables assumed with a fix value in simple-INSYDE were constant also in the original model (see e.g. IH, BH, GL). What is the difference compare to INSYDE? What are the fixed values adopted for simple-INSYDE?*

In INSYDE, the user can choose the value of the variables or use the default values, and this is valid also for variables as IH, BH, GL. In simple-INSYDE, they are implicitly considered so it is not possible to change them. The fixed value adopted in simple-INSYDE are the default values of INSYDE.

*R3-C5: Apart from a contained complexity, are there advantages on using a fixed values for same variables that can be directly estimated from other required in any case by the simple version? I am thinking for example to IA or EP, which depend on FA that is still necessary for the application of simple-INSYDE. Also in this case, how did you define those default values? Are they based on observations or assumptions?*

We apologize if the answer will not be thorough, but we did not fully understand the request of the referee. Variables as IA or EP or IP refer to geometric features of the building that, if unknown, INSYDE computes by means of some functions of FA. To develop simple-INSYDE we did not define these values, but we decided to maintain the original functions of FA, to be coherent with the original model, lines 70-71(66-67).

*R3-C6: L51: footprint area in table 1 is indicated as FA, not A. Also check Table 2.*

Yes, we used A and not FA for the simplified model, but we see it is confusing. We added a note in Table 1. See also R2-C12.

*R3-C7: L53: wouldn't be more correct saying "flood affected storeys"?*

We used the term "exposed to flood" and not affected, because we can use the damage model for damage forecast, and the storeys are not yet affected, but exposed to the event.

*R3-C8: L55: wouldn't be more precise saying "independent hazardous variables"?*

We mean all the variables, that refer both to hazard and vulnerability.

*R3-C9: L62: I suggest providing more details on these values. This would help the possibility to, eventually, apply the model elsewhere.*

The variable which were not selected as independent variable were set at default values defined in INSYDE, to be coherent with the original model. We added a sentence to clarify the choices of the independent variables, lines 62-63 (58-59). These choices are strongly related to data availability in

the region of interest and on assumptions about the typical building typologies, but, in another region and with another model, the criteria could be different. See also R2-C1.

*R3-C10: L73-76: honestly, these steps are not clear to me. If you can extend the explanation, or provide an example, it would be much easier to understand the procedure.*
We enriched the explanation but without getting too detailed in the description of the procedure for the definition of each function, in order to be coherent with the choice of a brief-communication, lines 82-89(76-82).

*R3-C11: Eq. 3): what happen in case LM is medium?*
LM medium is not considered in simple-INSYDE. The user must choose between low or high.

*R3-C12: Table 2: in order to make the model transferability easier it would be worth reporting in table 2 the range of the values considered for all the variables.*
For the variables that are not present in Table 2, the values are set as default, we added a sentence to clarify this, lines 92(85). See also R2-C3.

**List of all relevant changes made in the manuscript**

In the following, the list of the relevant changes made in the manuscript is supplied. The numbers of lines outside the brackets refer to the marked-up version of the manuscript, in brackets to the new version.

- Lines 20-21 (19-21): In the Introduction, we added a sentence to enrich the explanation of the limitations due to the complex structure of the original model.
- Lines 62-63 (58-59): We added a sentence to clarify the criteria of selection of independent variables in the simplified model version.
- Lines 77-79 (71-74): We enriched the explanation of the sensitivity analysis, without going into details to maintain the brief nature of the manuscript.
- Lines 83-89 (77-82): To make clearer the procedure of definition of the interpolating functions, we rephrased the explanation and added some examples.
- Lines 89-95 (82-88): We moved the lines about the calibration procedure before the equations and we added a sentence about the data sources of the probability distributions of the damage explicative variables.
- Table 2: We modified Table 2 to make it clearer using a synthetic notation for the probability distributions.
- Lines 121-128 (101-108): We moved some lines about calibration results after Table 2.
- Lines 155-171 (123-139): We moved part of the discussion after Figure 1.
- Lines 159-160 (127-128): We added a sentence in the discussion to underline the concept that the simplified model version does not significantly decrease the accuracy of the model.
- Lines 170-171 (138-139): We added a sentence to highlight the possibility to develop the proposed methodology for other multi-variable models as well.

Below, the marked-up version of the manuscript.

[revised manuscript text omitted]